# GAIA: ZERO-SHOT TALKING AVATAR GENERATION

**Tianyu He**[*], **Junliang Guo**[*], **Runyi Yu**[*], **Yuchi Wang**[*], **Jialiang Zhu, Kaikai An, Leyi Li**
**Xu Tan**[†]**, Chunyu Wang, Han Hu, HsiangTao Wu, Sheng Zhao, Jiang Bian**

Microsoft
{tianyuhe,junliangguo,v-runyiyu,v-yuchiwang,xuta}@microsoft.com

https://microsoft.github.io/GAIA

## ABSTRACT

Zero-shot talking avatar generation aims at synthesizing natural talking videos from speech and a single portrait image. Previous methods have relied on domain-specific heuristics such as warping-based motion representation and 3D Morphable Models, which limit the naturalness and diversity of the generated avatars. In this work, we introduce GAIA (Generative AI for Avatar), which eliminates the domain priors in talking avatar generation. In light of the observation that the speech only drives the motion of the avatar while the appearance of the avatar and the background typically remain the same throughout the entire video, we divide our approach into two stages: 1) disentangling each frame into motion and appearance representations; 2) generating motion sequences conditioned on the speech and reference portrait image. We collect a large-scale high-quality talking avatar dataset and train the model on it with different scales (up to 2B parameters). Experimental results verify the superiority, scalability, and flexibility of GAIA as 1) the resulting model beats previous baseline models in terms of naturalness, diversity, lip-sync quality, and visual quality; 2) the framework is scalable since larger models yield better results; 3) it is general and enables different applications like controllable talking avatar generation and text-instructed avatar generation.

## 1 INTRODUCTION

Talking avatar generation aims at synthesizing natural videos from speech, where the generated mouth shapes, expressions, and head poses should be in line with the speech content. Previous studies achieve high-quality results by imposing avatar-specific training (i.e., training or adapting a specific model for each avatar) (Thies et al., 2020; Tang et al., 2022; Du et al., 2023; Guo et al., 2021), or by leveraging template video during inference (Prajwal et al., 2020; Zhou et al., 2021; Shen et al., 2023; Zhong et al., 2023). More recently, significant efforts have been dedicated to designing and improving zero-shot talking avatar generation (Zhou et al., 2020; Wang et al., 2021a; Zhang et al., 2023b; Wang et al., 2023; Yu et al., 2022; Gururani et al., 2022; Stypułkowski et al., 2023), i.e., only a single portrait image of the target avatar is available to indicate the appearance of the target avatar. However, these methods relax the difficulty of the task by involving domain priors such as warping-based motion representation (Siarohin et al., 2019; Wang et al., 2021b), 3D Morphable Models (3DMMs) (Blanz & Vetter, 1999), etc. Although effective, the introduction of such heuristics hinders direct learning from data distribution and may lead to unnatural results and limited diversity.

In contrast, in this work, we introduce GAIA (Generative AI for Avatar), which eliminates the domain priors in talking avatar generation. GAIA reveals two key insights: 1) the speech only drives the motion of the avatar, while the background and the appearance of the avatar typically remain the same throughout the entire video. Motivated by this, we disentangle the motion and appearance for each frame, where the appearance is shared between frames and the motion is unique to each frame. To predict motion from speech, we encode motion sequence into motion latent sequence and

---

[*]Equal contribution.
[†]Corresponding author: Xu Tan (xuta@microsoft.com).

predict the latent with a diffusion model conditioned on the input speech; 2) there exists enormous diversities in expressions and head poses when an individual is speaking the given content, which calls for a large-scale and diverse dataset. Therefore, we collect a high-quality talking avatar dataset that consists of 16K unique speakers with diverse ages, genders, skin types, and talking styles, to make the generation results natural and diverse.

More specifically, to disentangle the motion and appearance, we train a Variational AutoEncoder (VAE) consisting of two encoders (i.e., a motion encoder and an appearance encoder) and one decoder. During training, the input of the motion encoder is the facial landmarks (Wood et al., 2021) of the current frame, while the input of the appearance encoder is a frame that is randomly sampled within the current video clip. Based on the outputs of the two encoders, the decoder is optimized to reconstruct the current frame. After we obtain the well-trained VAE, we have the motion latent (i.e., the output of the motion encoder) for all the training data. Then, we train a diffusion model to predict the motion latent sequence conditioned on the speech and one randomly sampled frame within the video clip, which provides appearance information to the generation process. During inference, given the reference portrait image of the target avatar, the diffusion model takes it and an input speech sequence as the condition, and generates the motion latent sequence that is in line with the speech content. The generated motion latent sequence and the reference portrait image are then leveraged to synthesize the talking video output using the decoder of the VAE.

For the collected dataset, to enable the desired information can be learned from data, we propose several automated filtration policies to ensure the quality of the training data. We train both the VAE and the diffusion model on the filtered data. From the experimental results, we have three key conclusions: 1) GAIA is able to conduct zero-shot talking avatar generation with superior performance on naturalness, diversity, lip-sync quality, and visual quality. It surpasses all the baseline methods significantly according to our subjective evaluation; 2) we train the model with different scales, varying from 150M to 2B. The results demonstrate that the framework is scalable since larger models yield better results; 3) GAIA is a general and flexible framework that enables different applications including controllable talking avatar generation and text-instructed avatar generation.

## 2    RELATED WORKS

Speech-driven talking avatar generation enables synthesizing talking videos in sync with the input speech content. Early methods have been proposed to train or adapt a specific model for each avatar with a focus on overall realness (Thies et al., 2020; Lu et al., 2021), natural head poses (Zhou et al., 2021), high lip-sync quality (Lahiri et al., 2021) and emotional expression (Ji et al., 2021).

Despite significant advances made by these methods, the costs are high due to the avatar-specific training. This motivates zero-shot talking avatar generation, where only one portrait image of the target avatar is given. However, animating a single portrait image is not easy due to the limited information we have. MakeItTalk (Zhou et al., 2020) handled this by first predicting 3D landmark displacements from the speech input, then the predicted landmarks are transferred to a warping-based motion representation (Siarohin et al., 2019), which is employed to warp the reference image to the desired expression and pose. Burkov et al. (2020) achieved pose-identity disentanglement, but needs additional fine-tuning for the unseen identities. More recently, SadTalker (Zhang et al., 2023b) leveraged 3DMMs as an intermediate representation between the speech and the video, and proposed two modules to predict the expression coefficients of 3DMMs and head poses respectively. In general, the current solutions relax the difficulty of the task by involving domain priors like warping-based transformation (Zhou et al., 2020; Wang et al., 2021a; 2022; Liu et al., 2022; Drobyshev et al., 2022; Gururani et al., 2022), 3DMMs (Ren et al., 2021; Zhang et al., 2021; 2023b), etc. Although the introduction of these heuristics makes the modeling easier, they inevitably hinder the end-to-end learning from data distribution, leading to unnatural results and limited diversity. PC-AVS (Zhou et al., 2021) and PD-FGC (Wang et al., 2023) similarly introduced identity space and non-identity space by leveraging the identity labels. The authors employed contrastive learning to align the non-identity space and speech content space. Our method differs in three ways: 1) they need additional driving video. Instead, we generate the entire motion from the speech at the same time and also provide the option to control the head pose; 2) they use contrastive learning to align speech and visual motion. In contrast, we leverage diffusion models to predict motion from the speech; 3) our method does not need additional identity labels. As verified in experiments, our method results in natural and consistent motion, and flexible control for talking avatar generation.

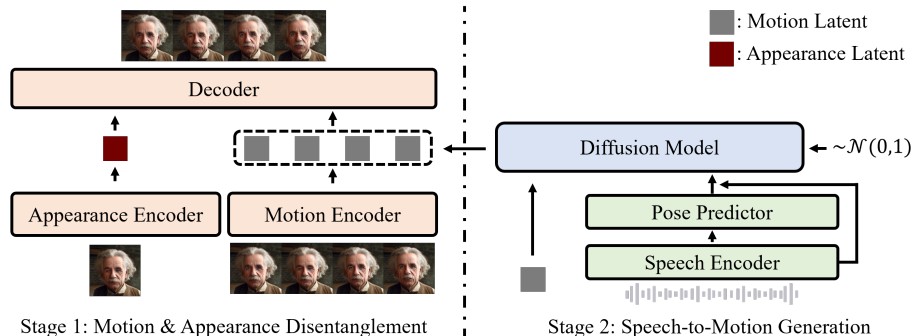

Stage 1: Motion & Appearance Disentanglement | Stage 2: Speech-to-Motion Generation

Figure 1: Method overview. GAIA consists of a VAE (the orange modules) and a diffusion model (the blue and green modules). The VAE is firstly trained to encode each video frame into a disentangled representation (i.e., motion and appearance representation) and reconstruct the original frame from the disentangled representation. Then the diffusion model is optimized to generate motion sequences conditioned on the speech sequences and a random frame within the video clip. During inference, the diffusion model takes an input speech sequence and the reference portrait image as the condition and yields the motion sequence, which is decoded to the video by leveraging the decoder of the VAE.

## 3 DATA COLLECTION AND FILTERING

A data-driven model is naturally scalable for large datasets, but it also requires high-quality data as it learns from data distribution. We construct our dataset from diverse sources. For high-quality public datasets, we collect High-Definition Talking Face Dataset (HDTF) (Zhang et al., 2021) and Casual Conversation datasets v1&v2 (CC v1&v2) (Hazirbas et al., 2021; Porgali et al., 2023) which contain thousands of identities (IDs) with a diverse set of ages, genders, and apparent skin types. In addition to these three datasets, we also collect a large-scale internal talking avatar dataset which consists of

Table 1: Statistics of the collected dataset.

| Datasets | Raw | | Filtered | |
|---|---|---|---|---|
| | #IDs | #Hours | #IDs | #Hours |
| HDTF | 362 | 16 | 359 | 14 |
| CC v1 | 3,011 | 750 | 2,957 | 330 |
| CC v2 | 5,567 | 440 | 4,646 | 183 |
| Internal | 8,007 | 7,000 | 8,007 | 642 |
| Total | 16,947 | 8,206 | 15,969 | 1,169 |

7K hours of videos and 8K unique speaker IDs, to make the resulting model scalable and unbiased. The overview of the dataset statistics is demonstrated in Tab. 1.

However, the raw videos are surrounded by noisy cases that are harmful to the model training, such as non-speaking clips and rapid head moves. To enable the desired information can be learned from data, we develop several automated filtration policies to improve the quality of the training data: 1) to make the lip motion visible, the frontal orientation of the avatar should be toward the camera; 2) to ensure the stability, the facial movement in a video clip should be smooth without rapid shaking; 3) to filter out corner cases where the lip movements and speech are not aligned, the frames that the avatar wear masks or keep silent should be removed. Please refer to Appendix A.1 for more details. After filtration, we find that a majority of raw videos are dropped, which is necessary for the training of a data-driven model according to our preliminary experimental results, where the video quality generated by models trained on raw videos falls behind the one trained on filtered data.

## 4 MODEL

### 4.1 MODEL OVERVIEW

The zero-shot scenario that generates a talking video of an unseen speaker with one portrait image and a speech clip requires two key capabilities of the model: 1) the disentangled representation of appearance and motion from the image, as the former should be consistent while the latter dynamic in the generated video; 2) generate the motion representation conditioned on the speech in each timestamp. Correspondingly, as shown in Fig. 1, we propose two models including a Variational AutoEncoder (VAE) (Kingma & Welling, 2014) that extracts image representations and a diffusion model for speech-to-motion generation.

**Problem Definition**   Given one portrait image $x$ and a sequence of speech clip $s = [s_1, ..., s_N]$, the model aims to generate a talking video clip $[x_1, ..., x_N]$ which is lip-syncing with speech $s$ and appearance consistent with image $x$.

## 4.2   MOTION AND APPEARANCE DISENTANGLEMENT

Given a frame of talking video $x$, we would like to encode its motion representation which will serve as the generation target of the diffusion model. Therefore, it is crucial to disentangle the motion and appearance representation from $x$. We propose a VAE that consists of two encoders, i.e., motion $\mathcal{E}_M$ and appearance encoder $\mathcal{E}_A$ and one decoder $\mathcal{D}$. We then use the appearance information from the $i$-th frame and the motion information from the $j$-th frame to reconstruct the $j$-th frame by the VAE, in order to prevent the leakage of the appearance information in reconstruction. In this way, as the $i$- and $j$-th frames from one video clip contain the same appearance but different motion information, i.e., the same person talking different words, the VAE model will learn to first extract the pure appearance feature from the $i$-th frame, and then combine it with the pure motion feature of the $j$-th frame to reconstruct the original $j$-th frame. The individuals of the $i$- and $j$-th frame can be flexibly chosen for both self-reconstruction and cross-reenactment settings.

**Motion and Appearance Encoder**   Specifically, denote the raw RGB image of $x$ as $x^a \in \mathbb{R}^{H \times W \times 3}$ and its landmark as $x^m \in \mathbb{R}^{H \times W \times 3}$ which is predicted by an external tool (Wood et al., 2021). The landmark is supposed to only contain the locations of key facial features such as the mouth, while the raw image provides other appearance information including identity and background. Given two frames $x(i)$ and $x(j)$ from one video clip, the model takes $x^a(i)$ and $x^m(j)$ as inputs to the appearance and motion encoder respectively, and produces their latent representations:

$$z^a(i) = \mathcal{E}_A(x^a(i)), \quad z^m(j) = \mathcal{E}_M(x^m(j)), \tag{1}$$

where $z^a(i) \in \mathbb{R}^{h^a \times w^a \times 3}$ and $z^m(j) \in \mathbb{R}^{h^m \times w^m \times 3}$. Note that in practice we use a smaller size of $h^m$ than $h^a$ as landmarks usually contain less information which is easier to encode. The two latent representations are then projected to the same size and concatenated together to reconstruct $x^a(j)$ by the decoder:

$$\hat{x}^a(j) = \mathcal{D}(z^a(i), z^m(j)). \tag{2}$$

The two encoders $\mathcal{E}_A$ and $\mathcal{E}_M$ share similar model architectures except for the downsampling factors, and $z^m(j)$ is first up-sampled to the same size as $z^a(j)$ followed by concatenation and projection and then served as the input to the decoder.

**Training**   We train the VAE model in an adversarial manner to learn perceptually rich representations following previous works (Esser et al., 2021; Rombach et al., 2022). In addition to the perceptual L1 reconstruction loss (Zhang et al., 2018) $L_{rec}(x, \hat{x})$ and the KL-penalty $L_{kl}(x)$ of the latent towards a standard normal distribution (Kingma & Welling, 2014), we introduce a discriminator $f_{dis}$ to distinguish between the real frame $x$ and the generated $\hat{x}$:

$$L_{dis}(x, \hat{x}) = \log f_{dis}(x) + \log(1 - f_{dis}(\hat{x})). \tag{3}$$

Then the total loss function of training the VAE can be written as:

$$L_{VAE} = \min_{\mathcal{E}_A, \mathcal{E}_M, \mathcal{D}} \max_{f_{dis}} (L_{rec}(x; \mathcal{E}_A, \mathcal{E}_M) + L_{kl}(x; \mathcal{E}_A, \mathcal{E}_M) + L_{dis}(x; f_{dis})). \tag{4}$$

## 4.3   SPEECH-TO-MOTION GENERATION

Once the VAE is trained, we are able to obtain a motion latent sequence $z^m \in \mathbb{R}^{N \times h^m \times w^m \times 3}$, an appearance latent sequence $z^a \in \mathbb{R}^{N \times h^a \times w^a \times 3}$ for each video clip. We also have its corresponding speech feature $z^s \in \mathbb{R}^{N \times d^s}$ extracted by wav2vec 2.0 (Baevski et al., 2020). We leverage a diffusion model with Conformer (Gulati et al., 2020) backbone $\mathcal{S}$ to predict the motion latent sequence $z^m$ conditioned on the paired speech feature $z^s$ and one reference frame $x(i)$. The speech feature gives the driving information and the reference frame provides identity-related information like facial contour, the shape of eyes, etc.

Since the speech feature $z^s$ comes from a fixed feature extractor (Baevski et al., 2020), to adapt it to our model, we process it with a lightweight speech encoder $\mathcal{A}$ before feeding it into the diffusion

model. Given that the diffusion model predicts the motion latent sequence, we thus use the motion latent $z^m(i)$ of the reference frame $x(i)$ as the condition, which is obtained by the pre-trained motion encoder $\mathcal{E}_M$. During training, the reference frame is randomly sampled within the video clip. Following previous practice (Du et al., 2023), we generate a pseudo-sentence for data augmentation by sampling a subsequence with a random starting point and a random length for each training pair.

**Diffusion Model**   Our goal is to construct a forward diffusion process and a reverse diffusion process that has a tractable form to generate data samples. The forward diffusion gradually perturbs data samples $z_0^m$ into Gaussian noise with infinite time steps. Then in the reverse diffusion, with the learned score function, the model is able to generate desired data samples $\hat{z}_0^m$ from Gaussian noise in an iterative denoising process. Formally, the forward diffusion can be modeled as the following stochastic differential equation (SDE) (Song et al., 2021):

$$\mathrm{d}z_t^m = -\frac{1}{2}\beta_t z_t^m \,\mathrm{d}t + \sqrt{\beta_t}\,\mathrm{d}w_t, \quad t \in [0, 1], \tag{5}$$

where noise schedule $\beta_t$ is a non-negative function, $w_t$ is the standard Wiener process (i.e., Brownian motion). According to previous literature (Song et al., 2021), the reverse diffusion that transforms the Gaussian noise to the data sample can therefore be written as:

$$\mathrm{d}z_t^m = -(\frac{1}{2}z_t^m + \nabla \log p_t(z_t^m))\beta_t \,\mathrm{d}t + \sqrt{\beta_t}\,\mathrm{d}\widetilde{w}_t, \quad t \in [0, 1], \tag{6}$$

where $\widetilde{w}_t$ is the reverse-time Wiener process, $p_t$ is the probability density function of $z_t^m$.

In addition, Song et al. (2021) have shown that there is an ordinary differential equation (ODE) for the reverse diffusion:

$$\mathrm{d}z_t^m = -\frac{1}{2}(z_t^m + \nabla \log p_t(z_t^m))\beta_t \,\mathrm{d}t. \tag{7}$$

Given the above formulation, we train a neural network $\mathcal{S}$ to estimate the gradient of the log-density of noisy data sample $\nabla \log p_t(z_t^m)$. As a result, we can model $p(z_0^m)$ by sampling $z_1^m \sim \mathcal{N}(0, 1)$ and then numerically solving either Equ. 6 or Equ. 7.

**Conditioning**   In addition to the noised data sample, our diffusion model processes additional conditional information: the noise time step $t$, the speech feature $z^s$, and a reference motion latent $z^m(i)$ coming from the same clip. Following previous successes (Ho et al., 2020; Rombach et al., 2022), the noise time step $t$ is projected to an embedding and then directly added to the input of each Conformer block. For the speech feature, since it should be aligned with the output, we add it to the hidden feature of each Conformer block in an element-wise manner. For the reference motion latent, we employ a cross-attention layer (Vaswani et al., 2017; Rombach et al., 2022) for each Conformer block, in which the hidden sequence in the Conformer layer acts as the query and the reference motion latent acts as the key and value.

**Pose-controllable Generation**   Predicting motion latent from the speech is a one-to-many mapping problem since there are multiple plausible head poses when speaking a sentence. To alleviate this ill-posed issue, we propose to incorporate pose information during training (Du et al., 2023; Tang et al., 2022). To achieve this, we extract the head poses $x^p \in \mathbb{R}^{N \times 3}$ (pitch, yaw, and roll) using an open-source tool [1], and add the extracted poses to the output of speech encoder $\mathcal{A}$ through a learned linear layer. By complementing the prediction with the head poses, the model puts more focus on generating realistic facial expressions, mouth shapes, etc.

To enable flexible generation during inference (i.e., one can use either the appointed head poses or the predicted one to control the generated talking video), we also train a pose predictor $\mathcal{P}$ to estimate the head poses according to the speech. The pose predictor $\mathcal{P}$ consists of several convolutional layers and is optimized by the mean square error between the extracted head poses $x^p$ and the estimated one $\hat{x}^p$.

**Training**   We jointly train the models $\mathcal{S}$, $\mathcal{A}$ and $\mathcal{P}$ with the following loss function:

$$L_{dif} = \mathbb{E}_{z_0^m, t}[||\hat{z}_0^m - z_0^m||_2^2 + L_{mse}(x^p, \hat{x}^p), \tag{8}$$

where the first term is the data loss, $\hat{z}_0^m = \mathcal{S}(z_t^m, t, z^s, z^m(i), x^p)$, and the second item is the loss for head pose prediction.

---

[1] `https://github.com/cleardusk/3DDFA`

Table 2: Quantitative comparisons of the GAIA VAE model with previous video-driven baselines.

| Methods | Self-Reconstruction | | | | | Cross-Reenactment | | |
|---|---|---|---|---|---|---|---|---|
| | FID↓ | LPIPS↓ | PSNR↑ | AKD↓ | MSI↑ | FID↓ | AKD↓ | MSI↑ |
| FOMM | 23.843 | 0.196 | 22.669 | 2.160 | 0.839 | 45.951 | 3.404 | 0.838 |
| HeadGAN | 21.499 | 0.278 | 18.555 | 2.990 | 0.835 | 90.746 | 5.964 | 0.788 |
| face-vid2vid | 18.604 | 0.184 | 23.681 | 2.195 | 0.813 | 28.093 | 3.630 | 0.853 |
| GAIA (Ours) | **15.730** | **0.167** | **23.942** | **1.442** | **0.856** | **15.200** | **2.003** | **1.102** |

Table 3: Quantitative comparisons of the GAIA framework with previous speech-driven methods. The subjective evaluation is rated at five grades (1-5) in terms of overall naturalness (Nat.), lip-sync quality (Lip.), motion jittering (Jit.), visual quality (Vis.), and motion diversity (Mot.). Note that, the Sync-D score for real video is $8.548$, which is close to ours.

| Methods | Subjective Evaluation | | | | | Objective Evaluation | | |
|---|---|---|---|---|---|---|---|---|
| | Nat.↑ | Lip.↑ | Jit.↑ | Vis.↑ | Mot.↑ | Sync-D↓ | MSI↑ | FID↓ |
| MakeItTalk | 2.148 | 2.161 | 1.739 | 2.789 | 2.571 | 9.932 | 1.140 | 28.894 |
| Audio2Head | 2.355 | 3.278 | 2.014 | 2.494 | _3.298_ | **8.508** | 0.635 | 28.607 |
| SadTalker | _2.884_ | _4.012_ | _4.020_ | _3.569_ | 2.625 | 8.606 | _1.165_ | **22.322** |
| GAIA (Ours) | **4.362** | **4.332** | **4.345** | **4.320** | **4.243** | _8.528_ | **1.181** | _22.924_ |

## 5 EXPERIMENTS

Benefitting from the disentanglement between motion and appearance, GAIA enables two common scenarios: the video-driven generation which aims to generate results with the appearance from a reference image and the motion from a driving video, and the speech-driven generation where the motion is predicted from a speech clip. The video-driven generation evaluates the VAE, while the speech-driven one evaluates the whole GAIA system. We compare GAIA with state-of-the-art methods for the two scenarios in Sec. 5.2, and further make detailed analyses in Sec. 5.3 to understand the model better. To verify the scalability of GAIA, we evaluate it at different scales in Sec. 5.3, i.e., from 150M to 2B model parameters in total. Due to the flexibility of our architecture, we also enable extended applications like text-instructed avatar generation, pose-controllable and fully controllable talking avatar generation (i.e., the mouth region is synced with the speech, while the rest of facial attributes can be controlled by the given talking video), which we demonstrate in Sec. 5.4.

### 5.1 EXPERIMENTAL SETUPS

**Datasets** We train our model on the union of the datasets described in Sec. 3, and we randomly sample 100 videos from them as the validation set. For the test set, to eliminate the potential overlap and evaluate the generality of our model, we create an out-domain test set by choosing 500 videos from TalkingHead-1KH (Wang et al., 2021b) dataset. We test all baselines on the same set.

**Implementation Details** We adjust the VAE and the diffusion model to different scales by changing the hidden size and the number of layers in each block, resulting in VAE of 80M, 700M, 1.7B parameters and diffusion model of 180M, 600M, 1.2B parameters. Refer to Appendix B.1 for the details of model architecture and training strategies.

**Evaluation** We utilize various metrics including subjective and objective ones to provide a thorough evaluation of the proposed framework. **Subjective metrics:** we conduct user studies to evaluate the lip-sync quality, visual quality, and head pose naturalness of the generated videos. 20 experienced users are invited to participate. We adopt MOS (Mean Opinion Score) as our metric. We present one video at a time and ask the participants to rate the presented video at five grades (1-5) in terms of overall naturalness, lip-sync quality, motion jittering, visual quality, and motion diversity respectively. **Objective metrics:** we adopt various objective metrics to evaluate the visual and motion quality of generation results. For visual quality, we report FID (Heusel et al., 2017) and LPIPS (Zhang et al., 2018) for perceptual similarity, and PSNR to measure the pixel-level mean squared error (MSE) between the ground truth and the reconstruction of the VAE. In addition, we detect the landmarks of

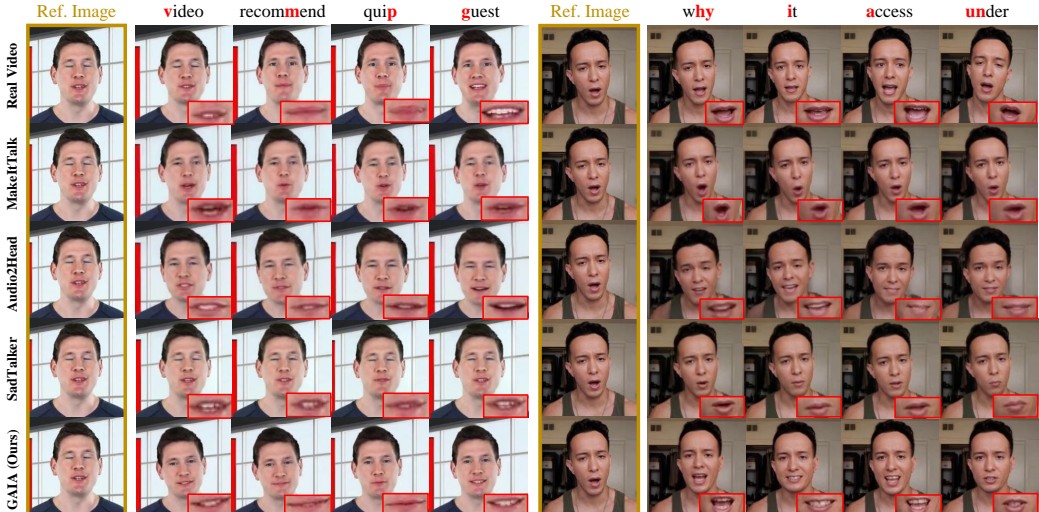

Figure 2: Qualitative comparison with the state-of-the-art speech-driven methods. It shows that GAIA achieves higher naturalness, lip-sync quality, visual quality and motion diversity. In contrast, the baselines tend to highly rely on the reference image (Ref. Image) therefore making generation with slight head motions (e.g., most of the baselines generate results with closed eyes when the eyes of the reference image are closed) or inaccurate lip synchronization.

ground truth and reconstructed images and report the Average keypoint distance (AKD) (Wang et al., 2021b) between them, to evaluate the motion quality of VAE reconstructions. Motion Stability Index (MSI) (Ling et al., 2022) which measures the motion stability of results is also reported. Following previous works (Thies et al., 2020; Tang et al., 2022), we adopt Sync-D (SyncNet Distance) to measure the lip-sync quality via SyncNet (Chung & Zisserman, 2016).

## 5.2 RESULTS

We compare the proposed GAIA model with state-of-the-art baselines in this section. Our model is general and can be applied to two common settings: the video-driven generation which aims to generate results with the appearance from a reference image and the motion from a driving video, and the speech-driven generation where the motion will be predicted from a speech clip. The video-driven generation evaluates the VAE, while the speech-driven one evaluates the whole GAIA system.

### 5.2.1 VIDEO-DRIVEN RESULTS

We consider two different settings of the video-driven talking avatar generation including self-reconstruction and cross-reenactment, depending on whether the individual of the appearance frame is consistent with the driving motion frames. Details of the two settings are provided in Appendix B.2. We compare with three strong baselines including FOMM (Siarohin et al., 2019), HeadGAN (Doukas et al., 2021) and face-vid2vid (Wang et al., 2021b), which are all equipped with feature warping, a commonly utilized prior technique in talking video generation. The results are shown in Tab. 2. The VAE of GAIA achieves consistent improvements over previous video-driven baselines, especially in the cross-reenactment settings, illustrating our model successfully disentangles the appearance and motion representation. Note that as a part of the data-driven framework, we try to make the VAE as simple as possible, and eliminate some commonly used external components such as a face recognition model (Deng et al., 2020) that provides identity-preserving losses.

### 5.2.2 SPEECH-DRIVEN RESULTS

The speech-driven talking avatar generation is enabled by predicting motion from the speech instead of the driving video. We provide both quantitative and qualitative comparisons with MakeItTalk (Zhou et al., 2020), Audio2Head (Wang et al., 2021a), and SadTalker (Zhang et al., 2023b) in Tab. 3 and Fig. 2. It can be observed that GAIA surpasses all the baselines by a large margin in terms of subjective evaluation. More specifically, as shown in Fig. 2, the baselines tend to make generation with high dependence on the reference image, even if the reference image is given with closed

Table 4: Scaling the VAE of GAIA. "#Params." and "#Hours" indicate the number of parameters and the size of the training dataset.

| #Params. VAE | #Hours | FID↓ |
|---|---|---|
| 80M | 0.5K | 18.353 |
| 80M | 1K | 17.486 |
| 700M | 1K | 15.730 |
| 1.7B | 1K | 15.886 |

Table 5: Scaling the diffusion model of GAIA. We use the VAE model of 700M parameters for all experiments.

| #Params. Diffusion | #Hours | Sync-D↓ |
|---|---|---|
| 180M | 0.1K | 9.145 |
| 180M | 1K | 8.913 |
| 600M | 1K | 8.603 |
| 1.2B | 1K | 8.528 |

eyes or unusual head poses. In contrast, GAIA is robust to various reference images and generates results with higher naturalness, lip-sync quality, visual quality and motion diversity. For the objective evaluation in Tab. 3, the best MSI score demonstrates that GAIA generates videos with great motion stability. The Sync-D score of $8.528$, which is close to the one of real video ($8.548$), illustrates that the generated videos have great lip synchronization. We obtain a comparable FID score to the baselines, which might be affected by the diverse head poses as we find that the model trained without diffusion realizes a better FID score in Tab. 6.

## 5.3 ABLATION STUDIES

### 5.3.1 ABLATION STUDIES ON SCALING

We change the scale of the model parameters as well as the training dataset to show the scalable of GAIA. For the model, we change the scales of VAE and Diffusion separately to study their influence on the framework. For the training set, we use the whole set with 1K hours or the subset of it.

The results are listed in Tab. 4 and Tab. 5, and we can find that scaling up the parameters and data size both benefit the proposed GAIA framework. For the VAE model, the results are tested with the self-reconstruction setting, which tends to converge when the model is larger than 700M. For the sake of efficiency, we utilize the 700M VAE model in our main experiments. As for the diffusion model, we still realize better results when the model grows up to 1.2B parameters.

### 5.3.2 ABLATION STUDIES ON PROPOSED TECHNIQUES

We study the proposed techniques in detail: 1) we encode each frame to the latent without disentanglement, and utilize the diffusion model to predict the latent (w/o disentanglement); 2) we generate the motion latent without making the condition on the head pose (w/o head pose); 3) we use the Conformer to predict the motion latent directly without the diffusion process (w/o diffusion); 4)

Table 6: Ablation studies on the proposed techniques.

| Methods | Sync-D↓ | MSI↑ | FID↓ |
|---|---|---|---|
| GAIA (700M + 180M) | 8.913 | 1.132 | 24.242 |
| w/o disentanglement | 12.680 | 1.423 | 140.009 |
| w/o head pose | 9.134 | 1.208 | 23.648 |
| w/o diffusion | 9.817 | 1.486 | 21.049 |
| w. landmark prediction | 9.331 | 1.038 | 27.022 |

we synthesize the coordinates of the landmarks, instead of the latent representation (w. landmark prediction). All experiments are conducted based on the 700M VAE model and the 180M diffusion model. As shown in Tab. 6, which demonstrates that: 1) the model without disentanglement fails to generate effective results; 2) the model trained without head pose or diffusion process yields inferior performance; 3) predicting landmarks, instead of the motion latent like ours, degrades the performance in all aspects. This illustrates that encoding motion into latent representation helps the learning of motion generation.

## 5.4 CONTROLLABLE GENERATION

**Pose-controllable Talking Avatar Generation**  As introduced in Sec. 4.3, in addition to predicting the head pose from the speech, we also enable the model with pose-controllable generation. We implement it by replacing the estimated head pose with either a handcrafted pose or the one extracted from another video, which is demonstrated in Fig. 3(a). Refer to Appendix D for more details.

**Fully Controllable Talking Avatar Generation**  Due to the controllability of the inverse diffusion process, we can control the arbitrary facial attributes by editing the landmarks during generation.

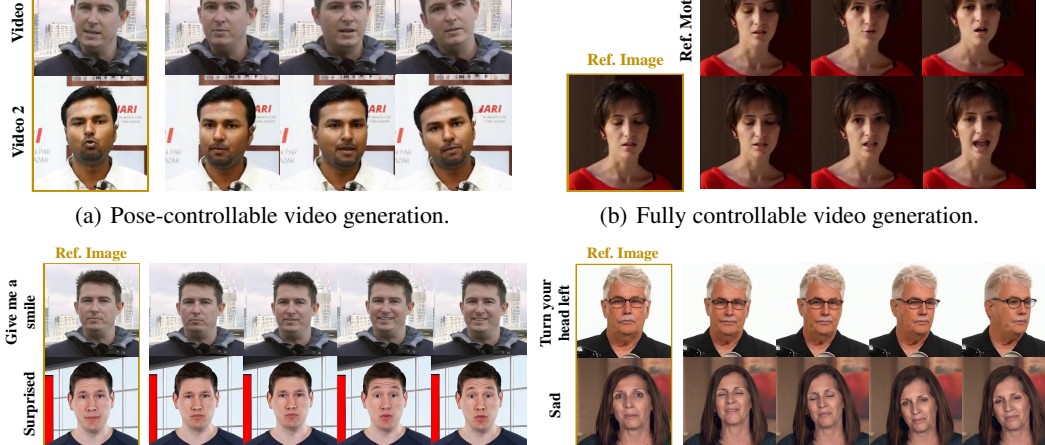

(a) Pose-controllable video generation.    (b) Fully controllable video generation.

(c) Text-driven video generation.

Figure 3: Examples of controllable and text-driven video generation. Due to the flexibility of our framework, 1) we enable multi-granularity motion control over the generated video. 2) we realize text-instructed video generation. See Sec. 5.4 for the details.

Specifically, we train a diffusion model to synthesize the coordinates of the facial landmarks. The landmarks that we want to edit are fixed to reference coordinates. Then we leave the model to generate the rest. In Fig. 3(b), we show the results of fully controllable generation, i.e., the mouth and jaw are synced with the speech, while the rest of the facial attributes are controlled by the reference motion. Refer to Appendix D for more details.

**Text-driven Video Generation**    In general, the diffusion model is a motion generator conditioned on speech, where the condition can be altered to other modalities flexibly. To show the generality of our framework, we consider textual instructions as the condition of the diffusion model, and enable the text-to-video generation (Fig. 3(c)). Refer to Appendix E.2 for more details.

## 5.5 Discussion

Different from previous works that employ warping-based motion representation (Wang et al., 2021a; Drobyshev et al., 2022), pre-defined 3DMM coefficients (Zhang et al., 2023b), we propose to eliminate these heuristics and generate the full motion latent at the same time. The framework discloses three insights: 1) the complete disentanglement between the motion and the appearance is the key to achieving zero-shot talking avatar generation; 2) handling one-to-many mapping with the diffusion model and learning full motion from real data distribution result in natural and diverse generations; 3) less dependence on heuristics and labels makes the method general and scalable.

## 6 Conclusion

We present GAIA, a data-driven framework for zero-shot talking avatar generation which consists of two modules: a variational autoencoder that disentangles and encodes the motion and appearance representations, and a diffusion model to predict the motion latent conditioned on the input speech. We collect a large-scale dataset and propose several filtering policies to enable the successful training of the framework end-to-end. The GAIA framework is general and scalable, which can provide natural and diverse results in zero-shot talking avatar generation, as well as being flexibly adapted to other applications including controllable talking avatar generation and text-driven video generation.

**Limitations and Future Works**    Our work still has limitations. For example, we leverage a pre-trained landmark extractor and a head pose extractor, which may hinder the end-to-end learning of the models. We leave the fully end-to-end learning (e.g., disentangle motion and appearance without the help of landmarks) as future work.

**Responsible AI Considerations**  GAIA is intended for advancing AI/ML research on talking avatar generation. We encourage users to use the model responsibly and to adhere to the Microsoft Responsible AI Principles [2]. We discourage users from using the method to generate intentionally deceptive or untrue content or for inauthentic activities. To prevent misuse, adding watermarks is a common way and has been widely studied in both research and industry works (Ramesh et al., 2022; Saharia et al., 2022). On the other hand, as an AIGC model, the generation results of our model can be utilized to construct artificial datasets and train discriminative models.

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

## A    DATA ENGINEERING

### A.1    DATA FILTRATION

As introduced in Sec. 3, we collected a large-scale talking avatar dataset which consists of 8.2K hours of videos and 16.9K unique speaker IDs. However, the raw videos are surrounded by noisy cases that are harmful to the model training, such as non-speaking clips and rapid head moves. To enable the desired information can be learned from data, we develop several automated filtration policies to improve the quality of the training data.

- To accurately learn the motion of the lip of the individual, it should be clearly visible by the model. Therefore, we maintain the frontal orientation of the individual toward the camera consistent in a video clip, and filter out frames with large deflections where the lips may be incomplete. Specifically, we calculate the clockwise angles formed by the positions of both eye corners in relation to the tip of the nose, using the tip of the nose as the horizontal reference line. Ideally, the angle should measure 180 degrees, for which we establish a range around it. Frames that fall outside this range will be dropped.
- To ensure the quality of the generation results, the facial movement in a video clip should be smooth without rapid shaking. Therefore, we monitor the face positions between adjacent frames and ensure that there is no significant displacement in continuous timestamps. We calculate the movement of the key point and face rectangle detected by an open-source detector[3], and limit the difference between two adjacent frames to a pre-defined threshold. In addition, we crop frames to place the talking head at the center to make its position consistent across different videos.
- To filter out corner cases where the lip movements and speech are not aligned, we detect and filter the frames where individuals are wearing masks or not speaking.

It is worth noting that the data requirements for the VAE and the diffusion model are different because the VAE model does not need to deal with the alignment between the speech and image, therefore we use a loose threshold for the filtration policies for training the VAE model.

We execute the filtration policies frame-by-frame for all raw videos, and retain the video segments with consecutive satisfactory frames longer than three seconds. The statistics of the filtered dataset are listed in Tab. 1. We can find that a majority of raw videos are dropped, which is necessary for the training of a data-driven model according to our preliminary experimental results, where the video quality generated by models trained on raw videos falls behind the one trained on filtered data.

### A.2    SPEECH PROCESSING

For each obtained video clip, we extract its speech and normalize the speech to a proper amplitude range. To reduce the background noise, we also apply a denoiser (Defossez et al., 2020) for each normalized speech clip. Since deep acoustic features have been found to be superior to traditional acoustic features like MFCC and mel-spectrogram (Baevski et al., 2020), following previous practice (Du et al., 2023), we leverage a pre-trained wav2vec 2.0 (Baevski et al., 2020) to extract the speech feature from the speech.

## B    EXPERIMENTAL SETTINGS

### B.1    IMPLEMENTATION DETAILS

The VAE consists of traditional convolutional residual blocks, with downsampling factors as 8 and 16 for appearance and motion encoder respectively. By changing the hidden size and number of layers in a block, we can control the size of the VAE model, and result in `small` (80M parameters, $d_{hidden} = 128$, $n_{layer} = 2$), `base` (700M parameters, $d_{hidden} = 256$, $n_{layer} = 4$), and `large` (1.7B parameters, $d_{hidden} = 512$, $n_{layer} = 8$) settings. The learning rate is set to $4.5 \times e^{-6}$ and keeps constant during training. We use Conformer (Gulati et al., 2020) as the backbone of the diffusion model. Similarly, we adjust the hidden size and the number of layers to obtain the speech-to-motion

---

[3]`https://github.com/davisking/dlib`

Table 7: Quantitative comparisons of the GAIA `small` VAE model (80M parameters) trained on the VoxCeleb2 (Chung et al., 2018) dataset with previous video-driven baselines.

| Methods | Self-Reconstruction | | | | | Cross-Reenactment | | |
|---|---|---|---|---|---|---|---|---|
| | FID↓ | LPIPS↓ | PSNR↑ | AKD↓ | MSI↑ | FID↓ | AKD↓ | MSI↑ |
| FOMM | 23.843 | 0.196 | 22.669 | 2.160 | 0.839 | 45.951 | 3.404 | 0.838 |
| HeadGAN | 21.499 | 0.278 | 18.555 | 2.990 | 0.835 | 90.746 | 5.964 | 0.788 |
| face-vid2vid | 18.604 | 0.184 | **23.681** | 2.195 | 0.813 | 28.093 | 3.630 | 0.853 |
| GAIA (VoxCeleb2) | **16.099** | **0.173** | 22.896 | **1.434** | **1.083** | **27.643** | **2.968** | **1.035** |

models with different scales: `small` (180M parameters, $d_{hidden} = 512$, $n_{layer} = 6$), `base` (600M parameters, $d_{hidden} = 1280$, $n_{layer} = 12$), and `large` (1.2B parameters, $d_{hidden} = 2048$, $n_{layer} = 12$). The learning rate starts from $1.0 \times e^{-4}$ and follows the inverse square root schedule. For both the VAE and the diffusion model, we adopt Adam (Kingma & Ba, 2015) optimizer and train our models on 16 V100 GPUs. We use the resolution of $256 \times 256$ for all the settings.

### B.2 SETTINGS FOR VIDEO-DRIVEN EXPERIMENTS

For the self-reconstruction setting, we choose the first frame of each video as the input to the appearance encoder, and the others as driving frames whose landmarks are extracted and fed to the motion encoder. We test on all frames in the test set.

For the cross-reenactment setting, we follow previous works (Zhang et al., 2023a) and randomly sample one frame from other videos as the appearance. To eliminate the effects of randomness, we run 5 rounds for each driving video. We generate 100 frames in each round for each video.

### B.3 SETTINGS FOR SPEECH-DRIVEN EXPERIMENTS

During training, we randomly sample training pairs with length $N$ from 125 to 250 to augment the training set for the speech-to-motion model (Du et al., 2023). For each test video, we use the first frame of each video as the reference image for all the methods.

## C MORE EXPERIMENTAL RESULTS

Due to the limited space in the main paper, we provide more experimental results for both video-driven and speech-driven settings in this section.

### C.1 MORE VIDEO-DRIVEN RESULTS

In addition to training our model on the dataset we proposed, we also train the `small` model (80M parameters) on the VoxCeleb2 (Chung et al., 2018) dataset which is utilized by previous baselines such as HeadGAN (Doukas et al., 2021) and face-vid2vid (Wang et al., 2021b) to provide fair comparisons with them. The results are listed in Tab. 7. When trained on the same dataset, GAIA still outperforms previous baselines on most metrics, showing the effectiveness of our model.

We provide qualitative results of video-driven self-reconstruction and cross-reenactment, and compare them with FOMM (Siarohin et al., 2019) and face-vid2vid (Wang et al., 2021b) in Fig. 4. For the self-reconstruction task which is relatively simple, both baselines and our model can achieve good results, while our model recovers more fine-grained details such as wrinkles and skin textures.

For the cross-reenactment setting, or cross-identity reenactment in other words, our model clearly outperforms baselines by dealing well with motion disentanglement and appearance reconstruction simultaneously.

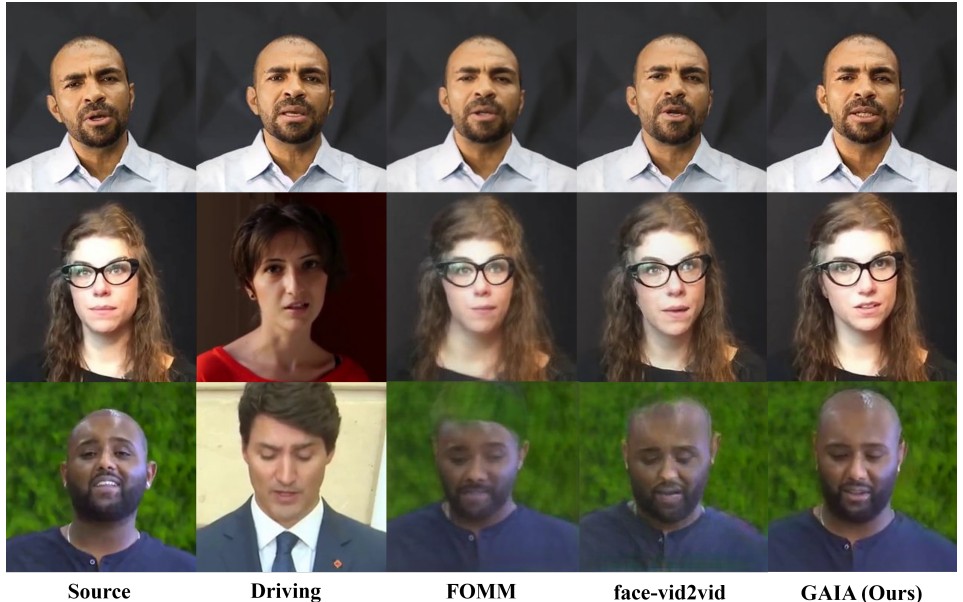

|  |  |  |  |  |
|:---:|:---:|:---:|:---:|:---:|
| **Source** | **Driving** | **FOMM** | **face-vid2vid** | **GAIA (Ours)** |

Figure 4: Qualitative examples of video-driven self-reconstruction (first row) and cross-reenactment (second and last rows) results from baselines and our GAIA VAE model.

Table 8: Quantitative comparisons of the GAIA framework with previous speech-driven methods. The subjective evaluation is rated at five grades (1-5) in terms of overall naturalness (Nat.), lip-sync quality (Lip.), motion jittering (Jit.), visual quality (Vis.), and motion diversity (Mot.). † the Sync-D score for real video is $8.548$, which is close to ours. * PD-FGC depends on extra driving videos to provide pose, expression and eye motions. We use the real (ground-truth) video as its driving video.

| Methods | Subjective Evaluation | | | | | Objective Evaluation | | |
|---|---|---|---|---|---|---|---|---|
|  | Nat.↑ | Lip.↑ | Jit.↑ | Vis.↑ | Mot.↑ | Sync-D↓ | MSI↑ | FID↓ |
| MakeItTalk | 2.148 | 2.161 | 1.739 | 2.789 | 2.571 | 9.932 | 1.140 | 28.894 |
| Audio2Head | 2.355 | 3.278 | 2.014 | 2.494 | _3.298_ | _8.508_ | 0.635 | 28.607 |
| PC-AVS | 2.797 | 3.843 | 3.546 | 3.452 | 2.091 | **8.341** | 0.677 | 59.464 |
| SadTalker | _2.884_ | _4.012_ | _4.020_ | _3.569_ | 2.625 | 8.606 | _1.165_ | **22.322** |
| PD-FGC* | 3.283 | 3.893 | 1.905 | 3.417 | 4.512 | 8.573 | 0.478 | 58.943 |
| GAIA (Ours) | **4.362** | **4.332** | **4.345** | **4.320** | **4.243** | $8.528^{†}$ | **1.181** | _22.924_ |

## C.2  MORE SPEECH-DRIVEN RESULTS

We provide full quantitative comparisons with MakeItTalk (Zhou et al., 2020), Audio2Head (Wang et al., 2021a), PC-AVS (Zhou et al., 2021), SadTalker (Zhang et al., 2023b), and PD-FGC (Wang et al., 2023) in Tab. 8. It can be observed that GAIA surpasses all the baselines by a large margin in terms of subjective evaluation. The best MSI score demonstrates that GAIA generates videos with great motion stability. The Sync-D score of $8.528$, which is close to the one of real video ($8.548$), illustrates that the generated videos have great lip synchronization.

We give more ablation studies for the proposed techniques in Tab. 9. All experiments are conducted based on the 700M VAE model and the 180M diffusion model. First, we study the conditioning mechanism for the speech-to-model generation: 1) we directly add both the speech feature and the reference motion latent $z^m(i)$ to each block of the Conformer layer (Spe. Add. & Ref. Add.); 2) in each cross-attention layer (Vaswani et al., 2017; Rombach et al., 2022), the hidden sequence in the Conformer layer acts as the query, and both the speech feature and the reference motion latent $z^m(i)$ are used as the key and value (Spe. Att. & Ref. Att.); 3) the speech feature is used as the key and value in each cross-attention layer, while the reference motion latent $z^m(i)$ is directly added to each block (Spe. Att. & Ref. Add.). We also replace the Conformer backbone in the diffusion model

Table 9: More ablation studies on the proposed techniques. See Sec. C.2 for details.

| Methods | Sync-D↓ | MSI↑ | FID↓ |
|---|---|---|---|
| GAIA (700M + 180M) | 8.913 | 1.132 | 24.242 |
| Spe. Add. & Ref. Add. | 8.989 | 1.125 | 28.189 |
| Spe. Att.  & Ref. Att. | 10.231 | 1.139 | 28.718 |
| Spe. Att.  & Ref. Add. | 10.180 | 1.015 | 24.021 |
| w/ Transformer | 9.312 | 0.623 | 25.385 |

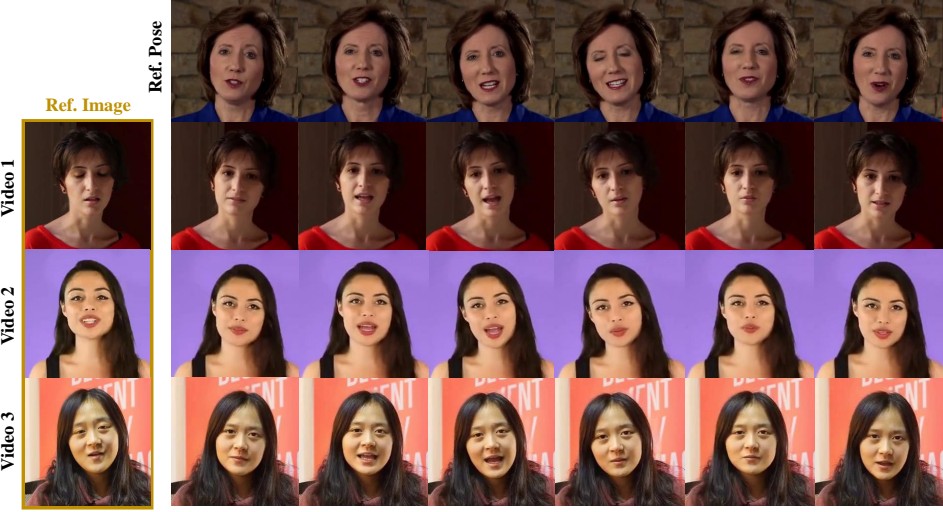

Figure 5: Examples of pose-controllable talking avatar generation. We extract the head poses from the reference video (first row), and use it to control the generation of different identities. Note that in this demonstration, we only control the head poses, while the lip motion and facial expression are generated according to the given speech, instead of the reference video.

with the Transformer (Vaswani et al., 2017) (w/ Transformer). From the table, we can observe that, adding the speech feature to the Conformer block and cross-attending to the reference motion latent (GAIA) achieves the best performance in terms of all three metrics. We also conclude that replacing the Conformer with the Transformer leads to significant motion jittering as the MSI score drops a lot.

# D  CONTROLLABLE TALKING AVATAR GENERATION

## D.1  POSE-CONTROLLABLE TALKING AVATAR GENERATION

As introduced in Sec. 4.3, in addition to predicting the head pose from the speech, we also enable the model with pose-controllable generation. We implement it by replacing the estimated head pose with either a handcrafted design pose or one extracted from another video. In detail, Fig. 3(a) is achieved by feeding the fixed pitch, yaw, and roll of head poses to the speech-to-motion model during generation. We also demonstrate the results of making generation with the head poses extracted from a reference video in Fig. 5. It can be observed that GAIA can generate results that head poses are consistent with the given one, while the lip motion is in line with the speech content.

## D.2  FULLY CONTROLLABLE TALKING AVATAR GENERATION

Due to the controllability of the inverse diffusion process, we can control the arbitrary facial attributes by editing the landmarks during generation. Specifically, we train a diffusion model to synthesize the coordinates of the facial landmarks. The landmarks that we want to edit are fixed to the given coordinates. Then we leave the model to generate the rest. This enables more flexible and fine-grained control over the generated videos. In particular, we provide the examples in Fig. 3(b), where all non-lip motion is aligned with the reference one, and the lip motion is in line with the speech content.

# E    TEXT-INSTRUCTED AVATAR GENERATION

## E.1    EXPERIMENTAL DETAILS

In general, the diffusion model is a motion generator conditioned on speech, where the condition can be altered to other modalities flexibly. To show the generality of our framework, we consider textual motion instructions as the condition of the diffusion model, to enable the text-instructed generation. Specifically, when provided with a single reference portrait image, the generation should follow textual instructions such as "please smile" or "turn your head left" to generate a video clip with the character performing the desired action.

We extract parallel data with text instructions and action videos from our dataset. We leverage the CC v1 dataset (Hazirbas et al., 2021) which contains data with off-screen instructional speeches and action videos of the participant. We then extract the instructional text and match it with the corresponding video clips of each action with the timestamp annotations. As a result, the text-instructed training set comprises of 28.8 hours videos and 24K textual instructional examples. We also modify the architecture of the diffusion model by substituting the speech feature with the textual semantic representations encoded by a pre-trained CLIP (Radford et al., 2021) text encoder.

## E.2    RESULTS

To evaluate the performance of the text-instructed generation, we randomly select 10 portraits that do not appear in the training set. For each of them, we provide 10 distinct textual instructions. Given the subjective nature of this task, we recruit 5 volunteers with relevant professional knowledge to rate the generation results between $0 - 5$ from three different perspectives: accuracy of instruction following, video quality, and identity preservation.

The three scores over the generated videos are $4.21$, $4.41$, and $4.64$ respectively, showing that the text-instructed model demonstrates strong abilities to generate actions that align with instructions, and the generated videos exhibit outstanding quality being natural and fluent. The text-instructed extension demonstrates the strong generality of the proposed GAIA framework.

