# OpenReview forum: "GAIA: Zero-shot Talking Avatar Generation"
_ICLR.cc/2024/Conference — ICLR 2024 poster_

### Official Review · Reviewer_tNS3 · 2023-10-30

**Soundness:** 3 good
**Presentation:** 3 good
**Contribution:** 3 good
**Rating:** 6
**Confidence:** 4

**Summary:**

This work aims to generate talking avatars with two separate modules: 1. first disentangle motion and appearance; 2 then generate head motions in accordance with speech.  As the proposed method does not utilize 3DMMs, the proposed data-driven method is promising to generate talking avatars with better diversity and naturalness.  The reported experiments show better quantitative results and better visual quality.

**Strengths:**

1. This work establishes a large-scale talking avatar dataset for data-driven talking avatar generation.

2. The proposed data-driven method could achieve taking avatar generation with superior performance on naturalness, diversity, lip-sync quality, and visual quality.

3. The proposed method is scalable, and the authors conduct experiments that show the larger model is employed, the better performance could be achieved.

4. The authors show that the proposed method could support many related applications, such as controllable talking avatar generation and text-driven video generation.

**Weaknesses:**

The authors did not provide a large number of generation examples. I hope it is possible to see more visual results.
For example,
1. long videos for the generated talking avatars;

2. different reference video/frame but the same driving video;

3. different driving video but the same reference video/frame.

**Questions:**

1. is it possible to drive cartoon characters (humanoid or non-humanoid)?

2. will the proposed method and the established dataset be publically available?

---

> ### Author Response · Authors · 2023-11-17
> **Author Response to Official Review by Reviewer tNS3**
>
> Dear Reviewer tNS3, thank you for taking the time to review. We appreciate that you found our model simple and scalable with superior performance.
>
> >**Q: The authors did not provide a large number of generation examples. I hope it is possible to see more visual results.**
>
> A: Thank you for your interest. We provide more samples on the anonymous **[demo page](https://gaiavatar.github.io/gaia/)** for convenience. More specifically, 1) an example of "long video generation" is provided in **[this video](https://gaiavatar.github.io/gaia/video/more/long_video.mp4)**. Since we focus on demonstrating the effectiveness of our proposed method, the long video generation is not yet well explored in the original manuscript. According to your comments, we present a simple yet effective strategy to generate infinite long videos -- by fixing the first several frames of each generation step during the reverse diffusion process. In this way, we obtain a smooth transition between each sub-video. 2) more examples of "different reference frame but the same driving video" are provided in **[this video](https://gaiavatar.github.io/gaia/video/more/different_reference_same_driving.mp4)**. 3) more examples of "different driving video but the same reference frame" are provided in **[this video](https://gaiavatar.github.io/gaia/video/more/different_driving_same_reference.mp4)**.
>
> We hope the above videos could make our results more clear.
>
> >**Q: Is it possible to drive cartoon characters (humanoid or non-humanoid)?**
>
> A: Sure. GAIA supports cartoon characters well. Examples can be found at **[this video](https://gaiavatar.github.io/gaia/video/more/cartoon1.mp4)**, **[this video](https://gaiavatar.github.io/gaia/video/more/cartoon2.mp4)** and also **[this video](https://gaiavatar.github.io/gaia/video/more/long_video.mp4)**, which demonstrates that our method generalizes well to out-of-domain data.
>
> >**Q: Will the proposed method and the established dataset be publically available?**
>
> A: We promise that the code will be released upon publication. For the dataset and pre-trained models, since privacy information is involved (i.e., real human faces), we are going through an internal compliance review and will try our best to mitigate the privacy risks and release them to facilitate future research.
>
> We welcome further discussion and are willing to answer any further questions.

---

> ### Author Response · Authors · 2023-11-21
> **We hope that our response addresses your concern**
>
> Dear Reviewer tNS3,
>
> We greatly appreciate the time you've invested in reviewing our response. Having submitted our rebuttal, we are eager to know if our response has addressed your concern. As the end of the rebuttal phase is approaching, we look forward to hearing from you for any further clarification that you might require.
>
> Best,
>
> Submission 5722 authors

---

### Official Review · Reviewer_Lkke · 2023-10-30

**Soundness:** 3 good
**Presentation:** 3 good
**Contribution:** 3 good
**Rating:** 6
**Confidence:** 3

**Summary:**

This paper proposes a data driven approach for generation of 2D avatars. The method disentangles motion and appearance and uses a diffusion model to allow motion generation conditioned on pose and speech data.

**Strengths:**

1) The method is conceptually simple and sensible.
2) It is shown to scale well in terms of model size and as a self-supervised method can utilize readily available training data at scale.
2) Method requires very few pretrained components.
3) Evaluation includes user study which is always good for addressing output quality.
4) Method is highly flexible and allows a high degree of control from pose, facial attributes and text.

**Weaknesses:**

1) A comparison with https://arxiv.org/pdf/2012.08261.pdf for video driven is critically missing as a recently proposed SOTA method. In their paper they show improvements compared to face-vid2vid and FOMM which are used as baselines here and they provide a pretrained checkpoint.

**Questions:**

1) will the dataset be shared as part of this submission?
2) It would be interesting to see an ablation on training data size to assess whether there are benefits from scaling data further.
3) How does randomness from the diffusion model affect generations?

---

> ### Author Response · Authors · 2023-11-17
> **Author Response to Official Review by Reviewer Lkke**
>
> Dear Reviewer Lkke, thank you for your positive and thoughtful feedback and for looking into every detail of our work.
>
> >**Q: The comparison with HeadGAN for the video-driven generation.**
>
> A: Thanks for reminding us of this important related work! We evaluate this model with the released checkpoint on our test set. To ensure fair comparisons, in addition to the model trained on our dataset, we also train one smaller GAIA model with 80M parameters on the VoxCeleb2 dataset to keep consistent with HeadGAN. We test models on the video-driven self-reconstruction and cross-reenactment settings, and the results are shown below.
>
> Table 5. Quantitative comparisons of self-reconstruction with HeadGAN.
>
> |                       |    FID$\downarrow$ |   LPIPS$\downarrow$   |   PSNR$\uparrow$ | AKD$\downarrow$  |   MSI$\uparrow$ |
> | --------------------- | ----- | ----- | ----- | ----- | ----- |
> | HeadGAN          | 21.499 | 0.278  | 18.555 | 2.990  | 0.835 |
> | GAIA-small w/ VoxCeleb2 | 16.099 | 0.173  | 22.896 | 1.434  | 1.083 |
> | GAIA | 15.730 | 0.167  | 23.942 | 1.442 | 0.856 |
>
> Table 6. Quantitative comparisons of cross-reenactment with HeadGAN.
>
> |                       |  FID$\downarrow$ | AKD$\downarrow$   |  MSI$\uparrow$  |
> | --------------------- | ----- | ----- | ----- |
> | HeadGAN    | 90.746 | 5.964  | 0.788 |
> | GAIA-small w/ VoxCeleb2 | 27.643 | 2.968  | 1.035 |
> | GAIA | 15.200 | 2.003  | 1.102
>
> From the results, GAIA achieves consistent improvements over HeadGAN across different metrics and settings. We have added the comparison with HeadGAN to Table 2 and 7 in the manuscript.
>
> >**Q: Will the dataset be shared as part of this submission?**
>
> A: We promise that the code will be released upon publication. For the dataset and pre-trained models, since privacy information is involved (i.e., real human faces), we are going through an internal compliance review and will try our best to mitigate the privacy risks and release them to facilitate future research.
>
>
> >**Q: The ablation on training data size.**
>
> A: We have investigated the scaling of training data in Tables 4 \& 5 of the manuscript, where \#Hours indicates the total training size. It can be observed that, with the same model size (80M VAE and 180M diffusion), our method obtains better results on more training data (e.g., 17.486 vs. 18.353 FID score, 8.913 vs. 9.145 Sync-D score). It is also interesting to see what will happen on larger data size (i.e., more than 1K hours). We leave it as future work.
>
>
> >**Q: How does randomness from the diffusion model affect generations?**
>
> A: Thank you for your interest! To show the effect of randomness, we conduct three kinds of visualizations in **[this video](https://gaiavatar.github.io/gaia/video/more/randomness.mp4)**: 1) we use different random seeds (i.e., seed 42 \& 82) for the generation, from which we observe that different seeds make slight differences for the generated videos; 2) we evaluate the models from different training epochs (i.e., epoch 1599 \& 1799), which shows the large difference on the generated motion; 3) we make generation with different reference images, which demonstrates that the reference image affects the appearance of the generated video since the missing information should be predicted by the model.
>
> We welcome further discussion and are willing to answer any further questions.

---

> > ### Public Comment · ~Istvan_Ketyko1 · 2024-05-13
> > **Code release date**
> >
> > Dear Authors,
> >
> > When is the code release expected?
> > Many thanks.

---

> > > ### Public Comment · ~Tianyu_He1 · 2024-05-14
> > > **Thanks for your attention**
> > >
> > > Hi,
> > >
> > > Thanks for your attention.
> > > As GAIA can generate photorealistic talking videos, to prevent unintended uses, we are going through the release process and internal compliance reviews with the legal department.
> > >
> > > Best,
> > > Authors

---

> ### Author Response · Authors · 2023-11-21
> **We hope that our response addresses your concern**
>
> Dear Reviewer Lkke,
>
> We greatly appreciate the time you've invested in reviewing our response. Having submitted our rebuttal, we are eager to know if our response has addressed your concern. As the end of the rebuttal phase is approaching, we look forward to hearing from you for any further clarification that you might require.
>
> Best,
>
> Submission 5722 authors

---

> > ### Comment · Reviewer_Lkke · 2023-11-22
> >
> > Dear Authors,
> >
> > Thank you for your response to my comments. The rebuttal has addressed my concerns and I am keeping my score as it is.

---

### Official Review · Reviewer_ft55 · 2023-10-31

**Soundness:** 3 good
**Presentation:** 3 good
**Contribution:** 2 fair
**Rating:** 6
**Confidence:** 5

**Summary:**

The paper proposes an audio-driven talking head synthesis model named GAIA that is an end-to-end trainable data-driven solution. The model has two main stages 1. disentanglement of motion and appearance with VAE 2. speech-to-motion generation based on diffusion model. Also, the paper proposes a new talking head dataset with 8.2K hours of video and 16.9K unique speakers.

**Strengths:**

The manuscript proposes a new dataset.

The writing is supported by equations and well-drawn figures that make the explanation clear.

Although the experiments with existing models are not enough (see weaknesses), the ablation study is rich and increases the overall quality.

**Weaknesses:**

The gap/ limitations of 3DMM-based models are found and addressed well by proposing an end-to-end trainable model. I am not sure it is novel enough as the other end-to-end trainable talking face synthesis models are not discussed enough.

The experiments are limited, especially comparison with end-to-end trainable models not provided. I suggest enriching the benchmarking with other existing models such as  PC-AVS and PD-FGC as they are also end-to-end trainable models.

Although the writing quality is decent, it is hard to follow as it refers to other sections frequently and other issues (see Questions 1 and 2).

**Questions:**

1. In section 3, what does 'we collect High Definition Talking Face Dataset (HDTF) (Zhang et al., 2021) and Casual Conversation datasets v1&v2 (CC v1&v2) (Hazirbas et al., 2021; Porgali et al., 2023)' and 'we also collect a large-scale internal talking avatar dataset named AVENA' mean? Does it mean you collect those datasets you use their sample in your dataset or you use their samples in your training but they are not in your dataset? From the supplementary material, my understanding is you collect AVENA and use samples from other datasets (HDTF, CC v1, and v2) in the training of your model. Could you please elaborate and make it clear?

2. Why do you have a discussion section at the end of Section 4 Model? I think it makes more sense in/after experiments. So, you can consider reorganizing the manuscript to have a better flow and complete discussion.

3. 3. I am not sure the model can be named as zero-shot as it requires one shot for unseen faces. So, could you elaborate on the following '... generates a talking video of an unseen speaker with one portrait image ...'?

3. Ethical consideration is left in Appendix F. However, for this study ethical consideration is important. So, you might consider putting it into the main manuscript to give necessary importance if possible.

---

> ### Author Response · Authors · 2023-11-17
> **Author Response to Official Review by Reviewer ft55 (1/3)**
>
> Dear Reviewer ft55, thank you for taking the time to review and propose promising extensions. We appreciate that you found our rich ablation study and the good writing. We address your concerns as follows.
>
> >**Q: Comparisons with other end-to-end trainable talking face synthesis models.**
>
> A: Thanks a lot for reminding the related end-to-end trainable works! We take PC-AVS [1] and PD-FGC [2] as examples to make more discussions. PC-AVS [1] and PD-FGC [2] similarly introduced identity space and non-identity space. The identity space is obtained by leveraging the identity labels, while the non-identity space is disentangled from the inputs through random data augmentation [3]. The authors employed contrastive learning to align the non-identity space and speech content space (except pose). However, our method differs in three ways: 1) they need additional driving video to provide motion information like head pose. In contrast, we generate the full motion from the speech at the same time and also provide the option to control the head pose; 2) they use contrastive learning to align speech and visual motion, which may lead to limited diversity due to the one-to-many mapping nature between the audio and visual motion. In contrast, we leverage diffusion models to predict motion from the speech; 3) their identity information is extracted by using identity labels while our method does not need additional labels.
>
> We further highlight our contributions as follows:
>
> * We propose a novel and sensible (as recognized by Reviewer 1S6P \& Lkke) framework that eliminates the heuristics and generates the full motion latent at the same time. The method reveals three key insights: 1) the complete disentanglement between the motion (speech-related) and the appearance (speech-agnostic) is the key to success; 2) handling one-to-many mapping with the diffusion model and learning full motion from real data distribution result in natural and diverse generations; 3) less dependence on heuristics and labels makes the method general and scalable.
>
> * We achieve superior performance on naturalness, diversity, lip-sync quality, and visual quality (as recognized by Reviewer 1S6P \& Lkke \& tNS3).
>
> * We, for the first time, verify the scalability of our method in talking avatar generation (as recognized by Reviewer Lkke \& tNS3).
>
> * The method is general and flexible to support many applications: speech-driven only, coarse-grained control (controlled by head pose), fine-grained control (fully controllable), and text-instructed avatar generation (as recognized by Reviewer 1S6P \& Lkke \& tNS3), which has never been explored in previous literature.
>
> We have added the discussions in the revised manuscript and marked the revisions in blue in Section 2.
>
> [1] Pose-Controllable Talking Face Generation by Implicitly Modularized Audio-Visual Representation. CVPR 2021.
>
> [2] Progressive Disentangled Representation Learning for Fine-Grained Controllable Talking Head Synthesis. CVPR 2023.
>
> [3] Neural Head Reenactment with Latent Pose Descriptors. CVPR 2020.
>
> Due to character limitations, please refer to the next comment for other questions.

---

> ### Author Response · Authors · 2023-11-17
> **Author Response to Official Review by Reviewer ft55 (2/3)**
>
> >**Q: The experiments are limited, especially comparison with end-to-end trainable models not provided. I suggest enriching the benchmarking with other existing models such as PC-AVS and PD-FGC as they are also end-to-end trainable models.**
>
> A: Thanks for your suggestion! Following your advice, we have evaluated PC-AVS [1] and PD-FGC [2] with their released models on the same test set. The results are listed in the table below. The visual comparison is provided in **[this video](https://gaiavatar.github.io/gaia/video/more/comparison_with_PC_PD.mp4)** and the full comparison is presented in **[this video](https://gaiavatar.github.io/gaia/video/more/1_speech-driven_add.mp4)**.
>
> Table 4. Quantitative comparisons of speech-driven generation with previous end-to-end trainable models.
>
> |                       |  Nat.$\uparrow$   |   Lip.$\uparrow$   |  Jit.$\uparrow$  |   Vis.$\uparrow$  |   Mot.$\uparrow$   |  Sync-D$\downarrow$   |  MSI$\uparrow$   |   FID$\downarrow$ |
> | --------------------- | ----- | ----- | ----- | ----- | ----- | ----- | ----- | ----- |
> | PC-AVS      |    2.797   |  3.843  |  3.546  |  3.452  |  2.091  |  8.341  |    0.677  |  59.464 |
> | PD-FGC*     |    3.283   |  3.893  |  1.905  |  3.417   | 4.512 |   8.573  |    0.478 |   58.943 |
> | GAIA        |    4.362   |  4.332  |  4.345  |  4.320  |  4.243  |  8.528   |   1.181  |  22.924 |
>
> In this table, \* indicates that PD-FGC depends on extra driving videos to provide pose, expression, and eye motions. We use the real (ground-truth) video as its driving video (therefore the score of motion diversity is high). We also note that the Sync-D score for real video is 8.548, which is close to ours. Overall, our method beats the previous end-to-end trainable models in terms of overall naturalness (Nat.), lip-sync quality (Lip.), motion jittering (Jit.), visual quality (Vis.), and motion diversity (Mot.). We have added the comparison in Table 8 in the revised version.
>
> [1] Pose-Controllable Talking Face Generation by Implicitly Modularized Audio-Visual Representation. CVPR 2021.
>
> [2] Progressive Disentangled Representation Learning for Fine-Grained Controllable Talking Head Synthesis. CVPR 2023.
>
>
> >**Q: Although the writing quality is decent, it is hard to follow as it refers to other sections frequently and other issues (see Questions 1 and 2).**
>
> A: Thanks for the suggestion! We have removed the unnecessary referrals to make the manuscript easy to follow. For example, we remove the referral in Section 5.3.1 (Ablation Studies on Scaling) and Section 5.5 (Discussion). In the revised version, apart from the referrals to the tables or figures, most referrals are referred to the Appendix for detailed illustrations.
>
>
> >**Q: About the collection of HDTF, CC v1\&v2 and AVENA datasets.**
>
> A: Thanks for your careful reading of our paper and supplementary materials, and sorry for the confusion. Your understanding is correct. AVENA is an internal dataset collected from our institution, and we utilize the union of AVENA, HDTF, CC v1 and v2 as the training set of our model. We have revised the description in Section 3 (Data Collection and Filtering) and Section 5.1 (Experimental Setups) for better understanding and marked it in blue.
>
>
> >**Q: About the discussion section.**
>
> A: Thanks for the suggestion! We have reorganized the flow in the revised manuscript and marked the revisions in blue. For example, we moved the improved discussion to the end of the experiments (Section 5.5).
>
>
> >**Q: I am not sure the model can be named as zero-shot as it requires one shot for unseen faces. So, could you elaborate on the following '... generates a talking video of an unseen speaker with one portrait image ...'?**
>
> A: The term "zero-shot" is potentially ambiguous in literature. In this paper, we use the zero-shot term due to the following reasons:
> 1) the method is "zero-shot" typically in the sense that no gradient updates are performed;
> 2) while inference, the difference between zero-/one-/few-shot lies in the number of utilized demonstrations [1], where a demonstration usually indicates an input-output pair of data. In our case, the provided unseen face is not a kind of demonstration, making the inference procedure zero-shot;
> 3) to be differentiable with the related works where a fine-tuning stage on the given reference portrait image is needed for the unseen identities [2,3,4].
>
> We are open to further discussions on this concern.
>
> [1] Language Models are Few-Shot Learners. NeurIPS 2020.
>
> [2] Few-Shot Adversarial Learning of Realistic Neural Talking Head Models. ICCV 2019.
>
> [3] Neural Head Reenactment with Latent Pose Descriptors. CVPR 2020.
>
> [4] Tune-A-Video: One-Shot Tuning of Image Diffusion Models for Text-to-Video Generation. ICCV 2023.
>
> Due to character limitations, please refer to the next comment for other questions.

---

> ### Author Response · Authors · 2023-11-17
> **Author Response to Official Review by Reviewer ft55 (3/3)**
>
> >**Q: About the location of Ethical consideration.**
>
> A: Thanks for the suggestion, and we agree that ethical considerations are crucial for the completeness of the paper. We have revised the content of the manuscript and put the ethical considerations in Section 6 (Conclusion).
>
> We hope our rebuttal and paper revision can address your concerns. We welcome further discussion and are willing to answer any further questions.

---

> ### Author Response · Authors · 2023-11-21
> **We hope that our response addresses your concern**
>
> Dear Reviewer ft55,
>
> We greatly appreciate the time you've invested in reviewing our response. Having submitted our rebuttal, we are eager to know if our response has addressed your concern. As the end of the rebuttal phase is approaching, we look forward to hearing from you for any further clarification that you might require.
>
> Best,
>
> Submission 5722 authors

---

> ### Comment · Reviewer_ft55 · 2023-11-22
>
> Dear Authors,
>
> Thank you for the rebuttal and addressing my concerns.
>
> I have read the responses carefully. And I think, with the revisions, the overall quality has been improved which led me to reconsider my score in the final review. I tend to increase my rating.

---

### Official Review · Reviewer_1S6P · 2023-11-01

**Soundness:** 3 good
**Presentation:** 3 good
**Contribution:** 3 good
**Rating:** 8
**Confidence:** 4

**Summary:**

The authors propose to use a generative latent diffusion model to address the problem of talking head synthesis from audio and a single photo. The pipeline consists of a variational autoencoder that encodes the video frames into appearance and motion latent representations and a diffusion model that is trained to predict the pre-trained motion latent from audio and pose conditioning. The authors also propose to use a data filtering approach to remove the noisy samples from the dataset to achieve high-quality results. The experimental evaluation is quite extensive and includes audio, head pose, and text-driven examples.

UPD: The rebuttal has addressed my concerns, and the results look genuinely impressive.

**Strengths:**

- Impressive quality of the results in terms of both lipsync and visual quality
- The model's design is straightforward yet evidently effective
- The paper is well-written and the evaluation is pretty extensive

**Weaknesses:**

- Missing evaluation of disentanglement between appearance and pose latent codes, i.e., cross-reenactment with the motion codes extracted from the image of a different identity.
- Missing discussion of the related works, such as [1, 2], that explored the concept of pose-identity disentanglement for talking head synthesis before this work.
- As far as I can tell, the proposed method and the baselines were trained on different datasets. The resulting comparison evaluates the proposed framework _and_ the dataset at the same time. A comparison should include the experiments where base methods are trained on the same dataset, and the proposed method is trained on unprocessed datasets used in previous works.
- Comparison of the inference time between the compared methods is not provided. I would also argue that some baselines, such as SadTalker, can be substantially improved, given the computational budget of the proposed method that runs the diffusion model for every time step. Ex., with the fine-tuning of the model given the source frame.

[1] Burkov et al., "Neural Head Reenactment with Latent Pose Descriptors", CVPR 2020
[2] Drobyshev et al., "MegaPortraits: One-shot Megapixel Neural Head Avatars", ACMMM 2022

**Questions:**

- Please address the concerns mentioned in the weaknesses
- Could the authors clarify if they plan to release the filtered dataset and the pre-trained models?

---

> ### Author Response · Authors · 2023-11-17
> **Author Response to Official Review by Reviewer 1S6P (1/2)**
>
> Dear Reviewer 1S6P, we are grateful for your careful review and the valuable feedback that you provided for our paper. We appreciate that you found our paper well-written and sound with convincing results. We hope the following comments address your concerns.
>
> >**Q: Missing evaluation of disentanglement between appearance and pose latent codes, i.e., cross-reenactment.**
>
> A: We have evaluated the cross-reenactment performance of GAIA and baselines in the right half of Table 2 of the manuscript, denoted as "Cross-Reconstruction" in the original version, where GAIA achieves significant improvements over baselines. Sorry for the confusion about the notations we utilized, we have changed it to "Cross-Reenactment" in this version to keep consistent with baselines.
>
> >**Q: Missing discussion of some related works.**
>
> A: Thank you for pointing out the related works! We make discussions as follows. We have also added the discussions to Section 2 (Related Works) and the revisions are marked in blue.
>
> Latent Pose Descriptors [1] achieved pose-identity disentanglement, where the identity embedding is averaged across multiple frames and the pose embedding is obtained with augmented input. However, the model needs additional fine-tuning for the unseen identities while we can directly generalize to the unseen identities with only one reference portrait image.
>
> MegaPortraits [2] adopted warping-based transformation, which is similar to face-vid2vid [3], but uses latent descriptors to represent expression instead of keypoints like face-vid2vid [3]. However, as we have validated in the subjective comparison and the experiments of the original manuscript, the usage of the heuristics hinders direct learning from data distribution, leading to unnatural results and limited diversity.
>
> In addition, instead of supporting video-driven avatar generation only [1,2], GAIA contains both motion \& appearance disentanglement and speech-to-motion modules that support speech-driven, video-driven, pose-controllable, full-controllable and text-instructed avatar generation in a unified framework.
>
> [1] Burkov et al., "Neural Head Reenactment with Latent Pose Descriptors", CVPR 2020
>
> [2] Drobyshev et al., "MegaPortraits: One-shot Megapixel Neural Head Avatars", ACMMM 2022
>
> [3] Wang et al., "One-shot free-view neural talking-head synthesis for video conferencing", CVPR 2021
>
>
> >**Q: The proposed method and the baselines were trained on different datasets.**
>
> A: The reason we choose different datasets is that we find commonly utilized datasets such as VoxCeleb2 suffer a lot from the jittering content and thus have relatively low quality. However, we agree with the point that training on the same dataset provides fair comparisons. Therefore, we train our model on VoxCeleb2, the training dataset utilized by face-vid2vid, while keeping the test set unchanged. Due to the time limitation, we train a smaller model with 80M parameters and will train a larger one in the next. The results of video-driven self-reconstruction and cross-reenactment are listed as follows.
>
> Table 1. Quantitative comparisons of self-reconstruction when trained on VoxCeleb2 dataset.
>
> |                       |  FID$\downarrow$  | LPIPS$\downarrow$  | PSNR$\uparrow$ | AKD$\downarrow$   |  MSI$\uparrow$  |
> | --------------------- | ----- | ----- | ----- | ----- | ----- |
> | face-vid2vid    | 18.604 | 0.184  |  23.681 | 2.195  | 0.813 |
> | GAIA-small w/ VoxCeleb2 | 16.099 | 0.173  | 22.896 | 1.434  | 1.083 |
>
>
> Table 2. Quantitative comparisons of cross-reenactment when trained on VoxCeleb2 dataset.
>
> |                       |  FID$\downarrow$ | AKD$\downarrow$   |  MSI$\uparrow$  |
> | --------------------- | ----- | ----- | ----- |
> | face-vid2vid    | 28.093 | 3.630  | 0.853 |
> | GAIA-small w/ VoxCeleb2 | 27.643 | 2.968  | 1.035 |
>
> From the results, we can find that when trained on the same dataset, GAIA still outperforms face-vid2vid on most metrics except PSNR (though it is the smallest model), demonstrating the power of the proposed method. We have added the results in Table 7 and more discussions in Section C.1 (More Video-driven Results) of the manuscript.
>
> Due to character limitations, please refer to the next comment for other questions.

---

> ### Author Response · Authors · 2023-11-17
> **Author Response to Official Review by Reviewer 1S6P (2/2)**
>
> >**Q: Comparisons of the inference time are not provided.**
>
> A: Thank you for your interest! According to your suggestions, we have evaluated the inference time of the same speech (8s) for each method. Below are the results, where our diffusion step is set to 150 (the same as the one we used in all experiments).
>
> Table 3. Comparisons of the inference time of the same speech.
>
> |                       |  Inference Time |
> | --------------------- | ----- |
> | MakeItTalk                       |  12.398s |
> | Audio2Head                       |  4.878s |
> | SadTalker                         |  7.300s    |
> | PC-AVS                            | 2.202s  |
> | PD-FGC                            | 2.883s  |
> | GAIA (80M VAE + 180M Diffusion)  | 5.548s (2.639s + 2.909s) |
> | GAIA (700M VAE + 600M Diffusion)  | 20.473s (16.950s + 3.523s) |
>
> From the table above, we observe that our model achieves comparable inference time against the baselines (e.g., 2.639s for 80M VAE, 2.909s for 180M diffusion model). Although the larger models like 700M VAE and 600M diffusion need more inference time, it can be further reduced with advanced techniques (e.g., fast diffusion sampling, model distillation, etc.), yet is not the focus of this work. For the current studies, we have shown that our method is able to achieve substantial improvement over the state-of-the-art and illustrated the scalability and generalizability of the proposed method.
>
>
> >**Q: Could the authors clarify if they plan to release the filtered dataset and the pre-trained models?**
>
> A: We promise that the code will be released upon publication. For the dataset and pre-trained models, since privacy information is involved (i.e., real human faces), we are going through an internal compliance review and will try our best to mitigate the privacy risks and release them to facilitate future research.
>
> We welcome further discussion and are willing to answer any further questions.

---

> ### Author Response · Authors · 2023-11-21
> **We hope that our response addresses your concern**
>
> Dear Reviewer 1S6P,
>
> We greatly appreciate the time you've invested in reviewing our response. Having submitted our rebuttal, we are eager to know if our response has addressed your concern. As the end of the rebuttal phase is approaching, we look forward to hearing from you for any further clarification that you might require.
>
> Best,
>
> Submission 5722 authors

---

> > ### Comment · Reviewer_1S6P · 2023-11-22
> > **Thanks for the rebuttal**
> >
> > I thank the authors for the comprehensive rebuttal.
> >
> > In the inference time section, could you please clarify what the separate timings in the parenthesis denote?
> >
> > After reading the other reviews and the rebuttal, I am quite happy with the changes the authors have provided. I tend to improve my score in the final review.

---

> > > ### Author Response · Authors · 2023-11-22
> > > **Thanks for your reply**
> > >
> > > Dear Reviewer 1S6P,
> > >
> > > Thanks a lot for your engagement and positive comments! The separate timings in parentheses represent the VAE decoder (to generate frames from the latents) and Diffusion model (to generate motion latents from the speech), respectively. For example, 2.639s indicates the inference time for the VAE decoder, and 2.909s indicates the inference time for the diffusion model. Therefore, the whole model needs 2.639s+2.909s=5.548s in total. It can be observed that currently, the VAE is the inference bottleneck when scaled up, and we plan to optimize its architecture for efficiency in future work.
> > >
> > > We appreciate your satisfaction with the changes made and look forward to a potential improvement in our final review score.
> > >
> > >
> > > Best,
> > >
> > > Submission 5722 authors

---

### Author Response · Authors · 2023-11-17
**General Comments to All Reviewers**

We would like to thank all the reviewers for their time and effort in the review process. We appreciate that you found our work "sensible, scalable and controllable" (1S6P, Lkke, tNS3), "achieves impressive quality" (1S6P, Lkke, tNS3), "well-written and clear" (1S6P, ft55), and our experiments and evaluations "extensive and rich" (1S6P, ft55). Taking into account all the reviewers' comments, we've responded to each reviewer individually and uploaded a revised manuscript as well as demo videos in supplementary materials.
In addition, to make it more convenient to evaluate the results, we have built an anonymous **[demo page](https://gaiavatar.github.io/gaia/)** that consists of more visual results and comparisons. If you find our answers responsive to your concerns, we would be grateful if you would consider increasing your score.

Following the suggestions and comments of reviews, we have revised the manuscript and marked the revisions in blue, which include:

* We have added more discussions with related works and highlighted our contributions in Section 2 (Related Works). (Reviewer 1S6P, ft55)

* To make the experiments more convincing, we have added additional results on the VoxCeleb2 dataset in Table 7. (Reviewer 1S6P)

* We have added comparisons with other end-to-end trainable models such as PC-AVS and PD-FGC in Table 8. (Reviewer ft55)

* We have added comparisons with HeadGAN in Table 2 and 7. (Reviewer Lkke)

* We have replaced the term "cross-reconstruction" with "cross-reenactment" to keep consistent with baselines. (Reviewer 1S6P)

* We have clarified the description for the training data in Section 3 (Data Collection and Filtering) and Section 5.1 (Experimental Setups). (Reviewer ft55)

* We have moved the improved discussion to the end of the experiments in Section 5.5 and removed the unnecessary referrals. (Reviewer ft55)

* We have moved the ethical consideration from the appendix to the main content in Section 6 (Conclusion). (Reviewer ft55)

* We have reorganized the manuscript to make it more compact in Abstract and Section 1 (Introduction).

Please kindly check our updated manuscript. We welcome further discussion.

---

### Meta-Review · Area_Chair_kKhf · 2023-12-12

**Metareview:**

The submission proposes an audio-driven talking head synthesis model consisting of a variational autoencoder for disentangling videos into appearance and motion representations, and a diffusion model to predict motion from audio conditioning. All the reviewers agreed that the method was simple and the results provided were of higher quality than prior work. After the rebuttal, one reviewer increased their score from a 5 to 6, leading to a unanimous positive opinion of the work. The authors are recommended to use the reviews to improve their submission.

**Justification For Why Not Higher Score:**

The method improves over a previously existing line of works, and the method as such is not extremely novel or groundbreaking.

**Justification For Why Not Lower Score:**

Based on the reviews and the discussion with reviewers, there is no particular reason to overturn the 4 unanimous positive reviews.

---

### Decision · Program_Chairs · 2024-01-16

Accept (poster)